# Exploring leprosy perceptions in South Sulawesi, Indonesia: A mixed-methods study on knowledge, attitudes, practices, and stigma

Ralalicia Limato[1], Ida A. Sutrisni[1], Rahmat Sagara[1], Asyhad F. Abdillah[1], Yuliati Yuliati[2], Al Kadri[2], Sri V. Muchtar[3,4], Iqbal Elyazar[1], Hardyanto Soebono[5,6], Jennifer I. Van Nuil[7,8☯], Marlous L. Grijsen[1,8☯*]

1 Oxford University Clinical Research Unit Indonesia, Faculty of Medicine, Universitas Indonesia, Jakarta, Indonesia, 2 Perhimpunan Mandiri Kusta (PerMaTa), Makassar, Indonesia, 3 Department of Dermatology and Venereology, Faculty of Medicine, Universitas Muslim Indonesia, Makassar, Indonesia, 4 Department of Dermatology and Venereology, Ibnu Sina Yayasan Wakaf-UMI Hospital, Makassar, Indonesia, 5 Department of Dermatology and Venereology, Universitas Gadjah Mada, Yogyakarta, Indonesia, 6 Center for Tropical Medicine, Faculty of Medicine, Public Health and Nursing, Universitas Gadjah Mada, Yogyakarta, Indonesia, 7 Oxford University Clinical Research Unit, Ho Chi Minh, Viet Nam, 8 Centre for Tropical Medicine and Global Health, Nuffield Department of Medicine, University of Oxford, United Kingdom

☯ Co-senior authors who contributed equally to this work
* mgrijsen@oucru.org

## Abstract

Leprosy-related stigma has a profound impact on affected individuals and their communities. Embedded within the MetLep trial of adjunctive metformin for multibacillary leprosy, we explored knowledge and perceptions of leprosy in endemic communities in Sulawesi, Indonesia. We conducted a mixed-methods cross-sectional study using interview-administered questionnaires and qualitative methods. Quantitative data were collected using the 'NLR Perception Toolkit', including a knowledge-questionnaire, the EMIC-Community Stigma Scale (EMIC-CCS) and the Social Distance Scale (SDS). Semi-structured interviews (SSIs) and focus group discussions (FGDs) were conducted to gain in-depth insights into perceived stigma, fears related to leprosy, care-seeking behaviour and access to healthcare. Quantitative data were analyzed using descriptive and multivariate analyses, while qualitative data were analyzed using inductive coding and content analysis. A total of 402 participants completed the questionnaires: 75 affected by leprosy (18.7%), 126 close contacts (31.3%), 150 community members (37.3%), and 51 healthcare workers (12.7%). Most participants were female (75.6%), with a median age of 42.0 years (IQR 32.3-52.0). Twenty-five individuals participated in SSIs and FGDs. Mean knowledge-score was low (3.0; SD 2.0; range 0–9), particularly regarding leprosy symptoms, cause, and transmission. Misconceptions included hereditary transmission and transmission through animal feces, sexual contact, or intercourse during menstruation. Mean EMIC-CSS and SDS scores were 11.2 (7.1; 0–28) and 8.7 (4.1; 0–21), respectively, with the highest stigma levels among community members. Qualitative findings

**Data availability statement:** Data will be made available upon reasonable request. As the study areas are clearly mentioned in our paper, we do not want to put our interviewees at risk, even though the data are de-identified. Leprosy is a highly stigmatized disease in this community and we don't want to risk identification of our participants who were willing to share their personal stories and experiences. Point of contact: Oxford University Clinical Research Unit (OUCRU) Indonesia: info.id@oucru.org.

**Funding:** The project was financially supported by Otto Kranendonk Foundation, The Netherlands Society for Tropical Medicine and International Health (NVTG) received by MLG. The funder had no role in study design, data collection and analysis, decision to publish, or preparation of the manuscript. JIVN and MLG are supported by the Wellcome Africa Asia Programme Vietnam core grant (106680/Z/14/Z).

**Competing interests:** The authors have declared that no competing interests exist.

revealed that care seeking experiences and perceived stigma were complex and context specific. Leprosy was locally referred to as "kandala", a term with negative socio-cultural connotations, and was often associated with witchcraft, sinful behavior, and moral transgression. Persons affected by leprosy reported community rejection, social exclusion, and avoidance of physical contact. Institutional stigma persisted within national leprosy programs. Addressing leprosy stigma requires context-specific, stigma-sensitive educational and public health interventions, supported by sustained political commitment and adequate funding.

## Introduction

Leprosy represents a global health challenge with over 200,000 new cases reported annually, predominantly affecting disadvantaged populations in low- and middle-income countries (LMICs) [1]. Leprosy affects individuals in their most productive stage of life, compromising their ability to meet social and financial responsibilities, perpetuating the cycle of poverty for those affected.

Indonesia ranks third globally in annual leprosy incidence, with over 14,000 new cases reported in 2023 [2]. Among these, 6% presented with visible (Grade 2) disability, suggesting diagnostic delays [3]. Although the World Health Organization (WHO) declared leprosy eliminated in Indonesia in 2000, which led to downscaling of the leprosy control program, the number of newly detected cases has stagnated. In response to this, the Indonesian government recently acknowledged the need to re-prioritize investments in leprosy elimination efforts, as outlined in the national long-term development plan for 2025–2045 [4].

Leprosy-related stigma is among the oldest forms of disease-related stigma, stemming from beliefs that leprosy was a punishment or sin and that those affected should be excluded from society [5]. Such misconceptions have fueled deep-rooted fear of the disease throughout history.

Leprosy and its associated disabilities often have a negative impact on the physical, psychological, social, and economic well-being of those affected [6]. Stigma and discrimination are major barriers to implementing successful healthcare programs, adversely affecting health-seeking behavior and treatment adherence, prolonging transmission [7,8]. This undermines control efforts and hinders the attainment of the WHO Global Leprosy Strategy 2021–30 of "Towards Zero Leprosy" [9].

In this study, we explored the knowledge and perceptions towards leprosy among leprosy-endemic communities in Indonesia to gain a better understanding of the context-specific, bio-social aspects and lived experiences of persons affected by leprosy within their social contexts.

## Methods

### Ethics statement

Ethical approval was obtained from the University of Gadjah Mada, Oxford Tropical Research Ethics Committee and London School of Hygiene & Tropical Medicine

Ethics Committee. This study was conducted in accordance with the international standards of ICH-Good Clinical Practice and Declaration of Helsinki. Each participant, and if applicable, the legally acceptable surrogate, was informed about the purpose and procedures of the study. Data was collected anonymously after the consent form was signed. A modest financial incentive was offered to compensate participants for their travel costs and time based on local standards.

## Study design

This study was embedded within the MetLep Trial, a clinical trial of adjunctive metformin combined with multi-drug therapy (MDT) for multibacillary leprosy in Indonesia (NCT05243654) [10]. We used a mixed-methods approach. The quantitative component, consisting of interview-administered questionnaires, provided measurable evidence on knowledge, attitudes and behaviour related to leprosy. The qualitative component, through semi-structured interviews (SSIs) and focus group discussions (FGDs), offered deeper insights into people's experiences, perceptions and coping strategies. Integrating these findings allowed us to develop a richer and more nuanced understanding of leprosy-related stigma [7,11] (Fig 1).

## Study setting

This study was conducted in Gowa district, southeast of Makassar, South Sulawesi in Eastern Indonesia, between June 5 2023 and January 8 2024. The area is leprosy-endemic and serves as one of the study sites for the MetLep Trial. In 2023, Gowa district had a population of 799,999, of whom 92.7% were Muslim [12], predominantly from the Bugis-Makassar ethnic groups. Leprosy has been prevalent in South Sulawesi for centuries, leading to the establishment of leprosy colonies. The history of the Jongaya leprosy colony in Makassar dates back to the Dutch colonial period. In 1934, the Dutch government and the Kingdom of Gowa donated land to quarantine and isolate people affected by leprosy, reflecting fears of disease transmission [13]. The current incidence of leprosy in Gowa is 1 per 10,000 population [14]. We selected two sub-districts in Gowa, one urban and one rural, to capture the different community contexts.

## Study population and sampling methods

**Interview-administered questionnaires.** This study included four groups of participants: 1) persons affected by leprosy; 2) their close contacts; 3) community members; and 4) healthcare workers (HCWs) from community health

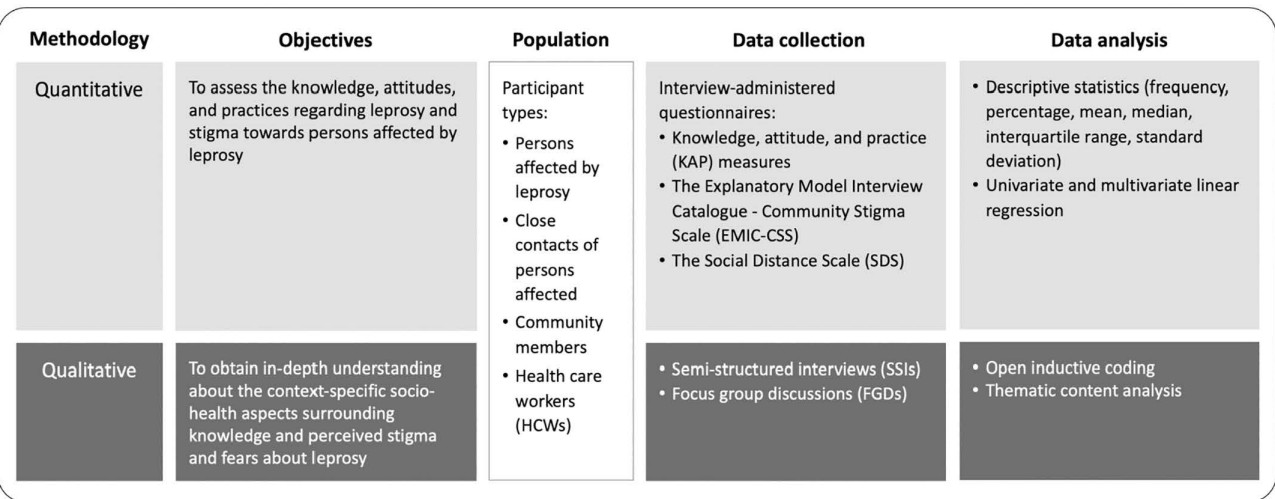

**Fig 1. A cross-sectional study using a mixed-methods approach.**

centers. Engaging these groups was essential to obtain a comprehensive understanding of the social dimensions of leprosy. Persons affected provided firsthand insights of lived experiences, including stigma, its impacts and barriers to care. Close contacts offered perspectives on transmission dynamics, as well as the stigma and fears associated with proximity to affected individuals. Community members contributed broader insights into societal attitudes, misconceptions, stigma, and discrimination. HCWs, who play a central role in diagnosis, treatment, health education, and stigma reduction, provided perspectives that highlight both strengths and gaps in leprosy control and care within the healthcare system.

Drawing on previous knowledge, attitudes, and practices (KAP) studies, we aimed to recruit a random sample of at least 400 participants [15]. We worked with two primary healthcare clinics, one clinic from each sub-district, to identify individuals who had been diagnosed with leprosy. Both clinics generated a list of 100 individuals who had been diagnosed with leprosy in previous five years; 40 people were then randomly selected from each list. In case selected individuals were not available or did not provide consent, we selected the next person on the list. Close contacts, community members and HCWs were recruited through convenience sampling, based on their availability and willingness during the study period.

**Semi-structured interviews and focus group discussions.** For the SSIs and FGDs, we aimed for a total of 20–25 participants. We aimed for a balanced representation in terms of age, sex, residence, and survey scores. We purposively selected and invited 4–5 survey participants from each of four participant groups who provided more elaborate information during the survey to participate in the SSIs or FGDs. We also identified relevant key informants who were not part of the survey to participate, i.e., religious leaders and persons affected by leprosy who actively engaged in a local leprosy organization. Participants invited for the SSIs were not the same as those for the FGDs.

## Eligibility criteria

A person affected by leprosy was eligible for the study if they had been diagnosed with leprosy. A close contact was any person listed by the persons affected as a close contact (e.g., household or family member, friend or neighbor) but was not diagnosed with leprosy themselves. A community member was any person living in the same village or neighborhood but was not classified as a close contact or an index case themselves. HCWs were health professionals working in primary healthcare clinics, with or without leprosy training or expertise. The key informants considered for the SSIs and FGDs were those who could provide rich information related to socio-cultural and religious contexts of leprosy. All participants were 18 years or older and willing and able to provide written informed consent.

## Data collection

**Interview-administered questionnaires.** Socio-demographic data were collected from all participants. The previously developed and locally translated and validated 'NLR Perception Study Toolkit' was used to collect data in REDCap among the four participant groups [11,15]. The Toolkit included a KAP questionnaire, the Explanatory Model Interview Catalogue Community Stigma Scale (EMIC-CSS) and the Social Distance Scale (SDS). The KAP questionnaire consisted of eight questions addressing the participants' knowledge towards leprosy, including questions on the cause, transmission and treatment of leprosy. Interviewers were trained not to lead participants to respond in a certain way and provide suggestive answers. The maximum KAP score that could be achieved was nine points, ranging from zero (poor knowledge) to 9 (high knowledge). Correct answers to a question were counted as one point in the absence of an incorrect answer. An additional seven questions were posed to individuals affected by leprosy, focusing on their personal experiences having leprosy and the manner in which they were treated by the community.

The EMIC-CSS and SDS questionnaires were aimed to assess perception and stigma towards the people affected by leprosy, and were therefore only administered to close contacts, community members, and HCWs. The EMIC-CSS measured perceived attitudes and behavior towards persons affected by leprosy. The scale consisted of 15 questions related

to areas of life that may be affected by stigma, such as concealment, avoidance, pity, shame, respect and marriage (prospects). The total score ranged from zero (no negative attitudes) to 30 (high negative attitudes). The SDS was designed to measure the social distance a participant would want to maintain towards a person affected by leprosy as a proxy for their attitudes. The interviewer first read a gender-specific vignette, a brief description describing the life of a person with leprosy, adapted to the local contexts. Following the vignette, participants answered seven questions about the individual described. The total score ranged from zero (indicating no negative attitudes) to 21 (most negative attitudes) [11,15].

**Semi-structured interviews and focus group discussions.** We conducted a preliminary analysis of the survey data before the SSIs and FGDs to refine and guide the selection of priority topics for deeper exploration. The SSIs and FGDs were conducted by two trained local interviewers focusing on the following areas: 1) knowledge and perceptions about leprosy; 2) openness and stigma towards persons affected by leprosy; 3) access to healthcare; 4) local beliefs regarding leprosy; and 5) perspectives on research and the MetLep trial.

A (de)briefing was conducted between RL and the local interviewers before and after each SSI and FGD. The SSIs and FGDs were conducted in Indonesian, audio-recorded with participants' permission, verbatim transcribed, and translated into English. Interviewers also prepared field notes during each session to capture non-verbal cues, such as facial expressions and body language. The SSIs ranged from 20 to 90 minutes, and the FGDs between 110 and 170 minutes. Pseudonyms were used for each interviewee to safeguard confidentiality and preserve a more personal narrative.

## Data analysis

**Quantitative data analysis.** Socio-demographic data were analyzed using descriptive analyses. The mean scores of all three scales were calculated per participant group. We performed chi-squared tests to assess the association between each independent variable and the participant groups, and Kruskal-Wallis chi-squared tests to determine whether the distribution of KAP, EMIC-CSS, and SDS scores were the same across all participant groups. For some questions in the KAP survey, multiple answers were permitted; the percentage of participants providing only the correct answer(s) in the absence of any incorrect response, was also calculated.

We developed a model to explain the linear relationship between the dependent and independent variables and used a Bayesian approach to estimate the parameters. In this analysis, the dependent variables included the KAP, EMIC-CCS, and SDS scores, while the independent variables, which were categorical data, included age, sex, marital status, education, occupation, monthly household income, area of residence, relationship with a person affected by leprosy, and participant group. The detailed description of the model and the categories of each independent variable can be accessed in the supporting document (S1 Text and Table A in S1 Text).

A univariate analysis was performed in which each independent variable was included one by one in the model for each dependent variable to estimate the intercept and slope parameters. Independent variables that significantly differed in their impact on the dependent variable were then included in a multivariate analysis. A backward stepwise regression was employed to refine the model. Four models for each dependent variable were made: i) the whole dataset involving all participant groups, ii) persons affected by leprosy, iii) close contacts combined with community members and iv) data from HCWs only. Drawing from previous studies, distinct models were developed for leprosy-affected individuals, because of their lived experiences and higher likelihood of knowing more about leprosy, and for HCWs, of whom majority was enrolled in tertiary education, which may have equipped them with specialized knowledge on leprosy [11]. The analyses were performed in R version 4.3.3 and RStudio version 2024.04.0.

**Qualitative data analysis.** For the qualitative analysis thematic content analysis was performed, which consisted of five steps: 1) data familiarization by (re-)reading the transcripts; 2) identification of initial themes and a thematic framework; 3) open inductive coding of the data; 4) charting, which involved rearranging the coded data based on the thematic framework; and 5) data mapping and interpretation [16]. To ensure consistency and reliability, four researchers

(RL, IAS, JIVN and MLG) read the transcripts and held debrief sessions to discuss the data and generate the themes. NVivo 12 was used for the qualitative data analysis.

## Results

Results include: 1) demographic information; 2) perceptions of leprosy, including cultural and religious contexts of leprosy, and leprosy cause, transmission, types, symptoms, and treatment; and 3) stigma, social distance, and impacts of leprosy.

### Demographic information

A total of 402 participants participated in the interview-administered questionnaires, with 75 persons affected by leprosy (18.7%), 126 close contacts (31.3%), 150 community members (37.3%), and 51 HCWs (12.7%), of whom 15 (29.4%) were involved in leprosy care. The median age of all participants was 42.0 years old (interquartile range [IQR] 32.2-52.0). The majority of participants were female (n = 304, 75.6%), Muslim (401, 99.8%), married (344, 85.6%), living in a rural setting (330, 82.1%) and working in the household (207, 51.5%). 29 participants (7.2%) had not received any form of education. 158 participants (39.3%) had a monthly household income below 95 USD per month. 76/327 participants (23.2%), excluding persons affected, had a relationship with someone affected by leprosy (Table 1).

Of the 75 leprosy-affected individuals, the majority were male (41, 54.7%) and diagnosed with multibacillary leprosy (52, 69.3%). The median duration of time since the person was diagnosed with leprosy was 42 months (IQR 26.5-61.8). Thirty-six individuals (48.0%) reported to have experienced a leprosy reaction; six individuals (8.0%) had developed visible (grade 2) disabilities. Thirty-nine (52.0%) mentioned that HCWs had explained the name of their condition as leprosy; 32 (42.7%) preferred the interviewer to use a different name for leprosy during the survey. We will explore facets of these perceptions in the sections that follow.

A total 25 people participated in SSIs and FGDs, 13 females (52%) and 12 males (48%), ranging in age from 18 to 72 years. Of these 25 people, 19 took part in the interview-administered questionnaires. The SSIs involved 15 participants: four persons affected by leprosy, four close contacts, four community members, and three HCWs – two of whom were directly involved in leprosy care. The two FGDs involved 10 people, each consisting of five persons affected by leprosy, with no overlap in participation between the SSIs and FGDs.

### Perceptions of Leprosy

**Knowledge, attitudes, and practices (KAP) scores.** The mean knowledge score (range 0–9) for all participants was 3.0 (standard deviation [SD] 2.0). HCWs had the highest scores (5.0, SD 1.7), followed by persons affected (3.4, 1.7), and close contacts (2.8, 1.7). Community members had the lowest knowledge scores (2.3, 1.9; $p < 001$). HCWs involved in leprosy care scored higher (5.5, 1.2) than those not involved in leprosy care (4.8, 1.9; $p = 0.1$). Only 21 of 402 participants (5.2%) scored >6 points of which the majority were HCWs (Table 1).

Multivariate analysis showed that no formal education, completing primary or secondary education, having an irregular income or a lower income (<95 USD/month) were associated with lower levels of leprosy knowledge. For close contacts and community members, not having a relationship with a person affected by leprosy was linked to less knowledge on the condition (Table 2).

**Leprosy in context: Kusta, kandala, wet and dry kusta, and skin and bone kusta.** In Indonesia, leprosy is commonly referred to as "kusta", a term derived from the Sanskrit word *kuṣṭha*, which broadly denotes a skin disease. Among the Bugis-Makassar, the predominant ethnic community living in South Sulawesi, *kusta* is also known as "kandala", which has a negative socio-cultural meaning. While participants had poor overall scores on the KAP survey, almost every interviewed participant knew about leprosy, or as termed in this context, *kandala. Kandala* was usually linked to the disabilities associated with leprosy.

**Table 1. Demographic characteristics and mean scores of interview-administered questionnaires of all participants and per participant group.**

| | Total | Person affected by leprosy | Close contacts | Community members | All HCWs | p-value between all groups | HCWs involved in leprosy care | HCWs not involved in leprosy care | p-value between HCWs |
|---|---|---|---|---|---|---|---|---|---|
| | N = 402 (%) | N = 75 (%) | N = 126 (%) | N = 150 (%) | N = 51 (%) | | N = 15 (%) | N = 36 (%) | |
| **Socio-demographic characteristics** | | | | | | | | | |
| **Age, median (IQR)** | 42.0 (32.2-52.0) | 49.0 (31.0-56.0) | 42.0 (33.2-51.0) | 43.0 (33.0-51.0) | 37.0 (30.5-45.5) | **0.03** | 40 (32.0-48.5) | 37 (30.0-45.0) | 0.40 |
| **Female** | 304 (75.6) | 34 (45.3) | 109 (86.5) | 115 (76.7) | 46 (90.2) | **<0.001** | 12 (80.0) | 34 (94.4) | 0.29 |
| **Education** | | | | | | | | | |
| No formal education | 29 (7.2) | 13 (17.3) | 11 (8.7) | 5 (3.3) | 0 (0.0) | **<0.001** | 0 (0.0) | 0 (0.0) | – |
| Primary school | 119 (29.6) | 24 (32.0) | 43 (34.1) | 52 (34.7) | 0 (0.0) | | 0 (0.0) | 0 (0.0) | |
| Secondary school | 196 (48.8)[a] | 36 (48.0) | 69 (54.8) | 88 (58.7) | 3 (5.9) | | 2 (13.3) | 1 (2.8) | |
| Tertiary education | 58 (14.4) | 2 (2.7) | 3 (2.4) | 5 (3.3) | 48 (94.1) | | 13 (86.7) | 35 (97.2) | |
| **Religion** | | | | | | | | | |
| Islam | 401 (99.8) | 75 (100.0) | 126 (100.0) | 150 (100.0) | 50 (98.0) | 0.08 | 14 (93.3) | 36 (100.0) | 0.65 |
| Christianity | 1 (0.2) | 0 (0.0) | 0 (0.0) | 0 (0.0) | 1 (2.0) | | 1 (6.7) | 0 (0.0) | |
| **Area of residence** | | | | | | | | | |
| Rural | 330 (82.1) | 67 (89.3) | 113 (89.7) | 122 (81.3) | 28 (54.9) | **<0.001** | 8 (53.3) | 20 (55.6) | 1.00 |
| Urban | 72 (17.9) | 8 (10.7) | 13 (10.3) | 28 (18.7) | 23 (45.1) | | 7 (46.7) | 16 (44.4) | |
| **Marital status** | | | | | | | | | |
| Married/Living together | 344 (85.6) | 55 (73.3) | 114 (90.5) | 129 (86.0) | 46 (90.2) | **0.03** | 14 (93.3) | 32 (88.9) | – |
| Separated/Divorced/Widowed | 19 (4.7) | 7 (9.3) | 4 (3.2) | 8 (5.3) | 0 (0.0) | | 0 (0.0) | 0 (0.0) | |
| Never married | 39 (9.7) | 13 (17.3) | 8 (6.3) | 13 (8.7) | 5 (9.8) | | 1 (6.7) | 4 (11.1) | |
| **Occupation** | | | | | | | | | |
| Health Care Worker | 51 (12.7) | 0 (0.0) | 0 (0.0) | 0 (0.0) | 51 (100.0) | **<0.001** | 15 (100.0) | 36 (100) | – |
| Self-Employee (merchant/business) | 43 (10.7) | 13 (17.3) | 13 (10.3) | 17 (11.3) | 0 (0.0) | | 0 (0.0) | 0 (0) | |
| Farmer | 27 (6.7) | 9 (12.0) | 4 (3.2) | 14 (9.3) | 0 (0.0) | | 0 (0.0) | 0 (0) | |
| Working in the Household | 207 (51.5) | 25 (33.3) | 90 (71.4) | 92 (61.3) | 0 (0.0) | | 0 (0.0) | 0 (0) | |
| Temporary Work/Unemployed | 74 (18.4) | 28 (37.3) | 19 (15.1) | 27 (18.0) | 0 (0.0) | | 0 (0.0) | 0 (0) | |
| **Monthly household income (USD)** | | | | | | | | | |
| Income <95 | 158 (39.3) | 39 (52) | 53 (42.1) | 55 (36.7) | 11 (21.6) | **<0.001** | 2 (13.3) | 9 (25.0) | 0.71 |
| Income 95–189 | 97 (24.1) | 16 (21.3) | 32 (25.4) | 43 (28.7) | 6 (11.8) | | 2 (13.3) | 4 (11.1) | |
| Income ≥190 | 75 (18.7) | 6 (8) | 12 (9.5) | 24 (16) | 33 (64.7) | | 11 (73.3) | 22 (61.1) | |
| Don't know/No Answer | 72 (17.9) | 14 (18.7) | 29 (23) | 28 (18.7) | 1 (2) | | 0 (0.0) | 1 (2.8) | |
| **Relationship with person affected by leprosy** | 111 (27.6) | 35 (46.7) | 71 (56.3) | 3 (2.0) | 2 (3.9) | **<0.001** | 1 (6.7) | 1 (2.8) | 1.00 |
| **Questionnaire scores**, mean (SD; range) | | | | | | | | | |
| Knowledge (KAP), range 0–9 | 3.0 (2.0; 0-9) | 3.4 (1.7; 0-8) | 2.8 (1.7; 0-7) | 2.3 (1.9; 0-9) | 5.0 (1.7; 1-8) | **<0.001** | 5.5 (1.2; 4-7) | 4.8 (1.9; 1-8) | 0.12 |
| Stigma (EMIC-CSS), range 0–30 | 11.2 (7.1; 0-28) | | 10.6 (7.7; 0-28) | 11.7 (6.8; 0-28) | 11.3 (6.2; 1-26) | 0.33 | 13.5 (6.7; 3-26) | 10.4 (5.8; 1-26) | 0.17 |
| Social distance (SDS), range 0–21 | 8.7 (4.1; 0-21) | | 8.5 (4.0; 0-20) | 9.5 (4.1; 0-21) | 7.3 (3.8; 0-16) | **0.006** | 5.7 (3.8; 0-13) | 8.0 (3.7; 2-16) | 0.07 |

*(Continued)*

**Table 1.** (Continued)

All data are listed as n (%), unless mentioned otherwise. Abbreviations: EMIC-CSS, Explanatory Model Interview Catalogue Community Stigma Scale; HCWs, Healthcare workers; KAP, knowledge, attitudes and practices; IQR, Interquartile range; SD, standard deviation; SDS, Social Distance Scale.

[a]78 (19.4%) and 118 (29.4%) participants had completed middle and secondary school, respectively; [b]For the multivariate regression analysis these groups were combined.

*"The term kandala in Bugis-Makassar means that the hands are clawing, are deformed, that is the understanding of the community, when they are thinking about the word kandala, they immediately think of people who have damaged clawed fingers..."* (FGD, Daeng (Dg.) Firzan, person affected by leprosy, male)

The term was also commonly used as a symbol of truthfulness, with the implication that individuals who were dishonest may develop leprosy-related disabilities. These disabilities associated with *kandala* subsequently became emblematic of punishment. For example, if an individual faced allegation of misconduct, they could assert their innocence by invoking the potential repercussions of *kandala*.

*"… and the word kandala as a curse that I will get leprosy when I commit a lie, for example, a sin, so that hurts people who have experienced leprosy. Kandala has been around for a long time, let's say it's like daily food, deeply embedded in Bugis-Makassar life."* (FGD, Dg. Firzan, person affected by leprosy, male)

Further, some people used *kandala* as a profanity, reinforcing the negative connotations linked to the term, and consequently the condition it represents.

*"Maybe it's because the posture of the body, they feel disgusted"* (FGD, Dg. Ngiji, person affected by leprosy, female).

Interestingly, not all people were familiar with the meaning of *kandala*, as mentioned by one of our participants, who was living in South Sulawesi but not part of the Bugis-Makassar community.

*"No. I heard people saying the word kandala, but I didn't know it was a disease like that"* (SSI, Dg. Tadi, close contact, female).

Some participants affected by leprosy distinguish between different forms of leprosy: "dry" denotes tuberculoid or paucibacillary leprosy, characterized by fewer skin lesions, low bacterial count and is therefore considered less contagious; and "wet" for lepromatous or multibacillary leprosy, presenting as a more severe, widespread form of disease with a higher bacterial load and greater infectivity.

*Participant: "If I am not mistaken, she said it [leprosy] was dry.*

*Interviewer:"Oh dry, but do you know the difference between dry or wet?"*

*Participant: "No."* (SSI, Dg. Sirajudding, community member, male)

Another participant whose late husband used to live in Jongaya (leprosy settlement) used a different explanation, which was leprosy of the skin for paucibacillary and leprosy of the bones for multibacillary leprosy.

*"My husband said there are two types of leprosy: of the skin and of the bones. Leprosy of the bones makes your bones break (fall off) … Having your bones affected is bad. They can break off without being cut off. This is not painful. If the*

**Table 2. The association between leprosy-related knowledge (KAP), community stigma (EMIC-CSS), social distance (SDS), and demographic characteristics for all participants and per participant group.**

| Data | Variables | Determinants of lower KAP scores | DIC | Determinants of higher EMIC-CSS scores | DIC | Determinants of higher SDS scores | DIC |
|---|---|---|---|---|---|---|---|
| Whole dataset | Age | – | -26.927 | age < 30 or ≥50 | 478.884 | age < 30 | 547.774 |
| | Marriage | – | | – | | – | |
| | Education | no formal education; primary education; secondary education | | secondary education; university | | – | |
| | Occupation | – | | self-employee; temporary worker/unemployed | | health care worker; self-employee; temporary worker; farmer | |
| | Gender | – | | – | | female | |
| | Relationship with person affected by leprosy | – | | – | | – | |
| | Monthly household income (USD) | irregular income; lower income (<95) | | moderate income (95–189); higher income (≥190) | | irregular income or lower income (<95); moderate income (95–189) | |
| | Area | – | | – | | – | |
| | Group | – | | health care worker or community | | community | |
| Healthcare worker group | Age | age < 50 | -35.405 | age 30–49; age ≥ 50 | 35.728 | – | 68.216 |
| | Marriage | – | | – | | – | |
| | Education | – | | secondary education | | – | |
| | Gender | – | | – | | female | |
| | Relationship with person affected by leprosy | – | | no relationship with person affected by leprosy | | – | |
| | Monthly household income (USD) | lower (<95) or moderate income (95–189) | | irregular income | | lower income (income <95) | |
| | Area | – | | – | | non-urban | |
| Close contacts and community group | Age | – | -66.704 | age < 30 | 429.422 | age < 30 | 440.944 |
| | Marriage | – | | – | | – | |
| | Education | no formal education; primary education; secondary education | | Secondary education; university | | no formal education | |
| | Occupation | self-employee or temporary work/unemployed | | self-employee; temporary worker/unemployed | | self-employee; temporary worker; farmer | |
| | Gender | – | | – | | female | |
| | Relationship with person affected by leprosy | no relationship with person affected by leprosy | | no relationship with person affected by leprosy | | – | |
| | Monthly household income (USD) | irregular income or lower income (<95) | | moderate income (95–189) | | – | |
| | Area | urban | | – | | urban | |
| | Group | – | | – | | – | |

*(Continued)*

| Data | Variables | Determinants of lower KAP scores | DIC | Determinants of higher EMIC-CSS scores | DIC | Determinants of higher SDS scores | DIC |
|---|---|---|---|---|---|---|---|
| Persons affected by leprosy group | Age | – | -28.268 | | | | |
| | Marriage | – | | | | | |
| | Education | no formal education; primary education; or university | | | | | |
| | Occupation | household worker; temporary worker/ unemployed | | | | | |
| | Gender | – | | | | | |
| | Relationship with person affected by leprosy | – | | | | | |
| | Monthly household income (USD) | – | | | | | |
| | Area | – | | | | | |

Abbreviations: DIC, Deviance Information Criterion; EMIC-CSS, Explanatory Model Interview Catalogue Community Stigma Scale; KAP, knowledge, attitudes, and practices; SDS, Social Distance Scale.

*skin is affected, you are fortunate because you can recover back to normal, but if treated too late, surely it can cause disability."* (SSI, Dg. Baji, person affected by leprosy, female)

**Community narratives on transmission.** In the KAP survey, 8% (32/402) of participants provided a correct response about the cause of leprosy and only 2.5% (10/402) on its transmission routes (Table 3). Participants had different perspectives and beliefs about whether leprosy is hereditary, an infectious disease, or both, potentially shaped by observations within their families or community or their own experiences.

*"Some are hereditary, and some are not… Sometimes there is no person affected by leprosy in the family, but you still get it. My cousin got leprosy within his family because his mother was sick as well."* (SSI, Dg. Salehuddin, person affected by leprosy, male)

*"The HCW acknowledged leprosy is infectious, but I don't believe it… My first husband, from childhood until marriage, lived in Jongaya (leprosy settlement for people affected by leprosy in Makassar that was built during the Dutch colonial time). He was eating, drinking, and sleeping there, but he didn't show any skin symptoms. If leprosy is an infectious disease, the whole of Jongaya would have been affected by leprosy; in fact, the children are not sick even though their mothers are."* (SSI, Dg. Baji, person affected by leprosy, female)

Cultural and religious beliefs were also tied to the perceptions of leprosy, including the etiology and transmission. For example, 88 participants (21.9%) believed that leprosy was a punishment and caused by witchcraft or a sin, of whom 21 were HCWs (23.9%, Table 3). Persons affected by leprosy frequently consulted an *ustadz*, a religious figure well-versed in Islamic teachings, for guidance. According to Islamic beliefs, as cited by *ustadz*, leprosy may be the result of witchcraft. In this context, it is a supernatural power to harm other people, such as misfortune, illness, or even death.

**Table 3. Overview of KAP survey responses (the correct answers are shown in grey).**

| Topic | Responses to each question shown per group (N, %). Some questions permitted multiple answers, therefore percentages may exceed 100%. | Person affected by leprosy | Close contacts | Community members | All HCWs | HCWs involved in leprosy care | HCWs not involved in leprosy care | Percentage of participants providing correct answers only* |
|---|---|---|---|---|---|---|---|---|
| | | N=75 (%) | N=126 (%) | N=150 (%) | N=51 (%) | N=15 (%) | N=36 (%) | N=402 |
| Early symptoms | Loss of sensation | 17 (22.7) | 9 (7.1) | 8 (5.3) | 26 (51.0) | 8 (53.3) | 18 (50.0) | 21 (5.2%) |
| | Skin patches | 39 (52.0) | 31 (24.6) | 22 (14.7) | 25 (49.0) | 12 (80.0) | 13 (36.1) | |
| | Itchiness | 20 (26.7) | 26 (20.6) | 39 (26.0) | 21 (41.2) | 7 (46.7) | 14 (38.9) | |
| | Other (e.g., wounds on the skin, disabilities, tingling) | 29 (38.7) | 37 (29.4) | 45 (30.0) | 33 (64.7) | 10 (66.7) | 23 (63.9) | |
| | Don't know | 8 (10.7) | 60 (47.6) | 75 (50.0) | 3 (5.9) | 0 (0) | 3 (8.3) | |
| Cause of leprosy | Bacteria | 6 (8.0) | 9 (7.1) | 9 (6.0) | 17 (33.3) | 6 (40.0) | 11 (30.6) | 32 (8.0%) |
| | Unclean environment | 5 (6.7) | 3 (2.4) | 9 (6.0) | 12 (23.5) | 5 (33.3) | 7 (19.4) | |
| | Hereditary | 1 (1.3) | 22 (17.5) | 26 (20.6) | 12 (23.5) | 5 (33.3) | 7 (19.4) | |
| | Don't know | 49 (65.3) | 78 (61.9) | 96 (64.0) | 7 (13.7) | 0 (0) | 7 (19.4) | |
| | Other (e.g., punishment, sin, karma, witchcraft) | 18 (24.0) | 24 (19.0) | 25 (16.7) | 21 (41.2) | 7 (46.7) | 14 (38.9) | |
| Transmission | By air | 4 (5.3) | 6 (4.8) | 9 (6.0) | 8 (10.4) | 3 (20.0) | 5 (13.9) | 10 (2.5%) |
| | Skin contact | 7 (9.3) | 19 (15.1) | 39 (26.0) | 23 (45.1) | 10 (66.7) | 13 (36.1) | |
| | Sharing personal items, eating together, shaking hands | 5 (6.7) | 27 (21.4) | 26 (17.3) | 20 (39.2) | 2 (13.3) | 18 (50.0) | |
| | Sexual activity | 1 (1.3) | 7 (5.5) | 5 (3.3) | 3 (5.9) | 2 (13.3) | 1 (2.8) | |
| | Other (e.g., contaminated soil, insects, mosquitoes) | 7 (9.3) | 20 (15.9) | 15 (10.0) | 20 (39.2) | 7 (46.7) | 13 (36.1) | |
| | Don't know | 54 (72.0) | 72 (57.1) | 85 (56.7) | 3 (5.9) | 0 (0) | 3 (8.3) | |
| Is leprosy curable | Curable | 62 (82.7) | 94 (74.6) | 104 (69.3) | 49 (96.1) | 15 (100) | 34 (94.4) | 309 (76.9%) |
| | Not curable | 1 (1.3) | 4 (3.2) | 13 (8.7) | 0 (0) | 0 (0) | 0 (0) | |
| | Don't know | 12 (16.0) | 28 (22.2) | 33 (22.0) | 2 (3.9) | 0 (0) | 2 (5.6) | |
| How is it **treated** | With medicines | 60 (80.0) | 88 (69.8) | 75 (50.0) | 46 (90.2) | 15 (100) | 31 (86.1) | 244 (60.7%) |
| | Other (e.g., by avoiding taboo food, religious practices, herbal treatment) | 4 (5.3) | 19 (17.5) | 44 (29.3) | 13 (25.5) | 3 (20.0) | 010 (27.8) | |
| | Cannot be treated | 0 (0) | 3 (2.4) | 8 (5.3) | 0 (0) | 0 (0) | 0 (0) | |
| | Don't know | 11 (14.7) | 22 (17.5) | 34 (22.7) | 1 (2.0) | 0 (0) | 1 (2.8) | |
| Contagious after treatment initiation | No | 42 (56.0) | 67 (53.2) | 69 (46.0) | 33 (64.7) | 10 (66.7) | 23 (63.9) | 211 (52.5%) |
| | Yes | 8 (10.7) | 19 (15.1) | 24 (16.0) | 13 (25.5) | 5 (33.3) | 8 (22.2) | |
| | Don't know | 25 (33.3) | 40 (31.7) | 57 (38.0) | 5 (9.8) | 0 (0) | 5 (13.9) | |
| Can disabilities be prevented | Yes | 28 (37.3) | 55 (43.7) | 56 (37.3) | 39 (76.5) | 10 (66.7) | 29 (80.6) | 178 (44.3%) |
| | No | 8 (10.7) | 21 (16.7) | 19 (12.7) | 8 (15.7) | 4 (26.7) | 4 (11.1) | |
| | Don't know | 39 (52.0) | 50 (39.7) | 75 (50.0) | 4 (7.8) | 1 (6.7) | 3 (8.3) | |
| Can leprosy be prevented | Yes, with preventive medicines | 29 (38.7) | 40 (31.7) | 37 (24.7) | 31 (60.8) | 9 (60.0) | 22 (61.1) | 110 (27.4%) |
| | Yes, by avoiding contact with people affected by leprosy or by isolating persons affected | 3 (4.0) | 17 (13.5) | 24 (16.0) | 11 (21.6) | 7 (46.7) | 4 (11.1) | |
| | Yes, with herbs, by religious ceremony or other practices | 9 (12.0) | 25 (19.8) | 33 (22.0 | 16 (31.4) | 5 (33.3) | 11 (30.6) | |
| | No, cannot be prevented | 2 (2.7) | 6 (4.8) | 5 (3.3) | 1 (2.0) | 0 (0) | 1 (2.8) | |
| | Don't know | 34 (45.3) | 48 (38.1) | 67 (44.7) | 5 (9.8) | 0 (0) | 5 (13.9) | |

* Only participants who provided correct answers, in the absence of any incorrect answers, were counted. Abbreviations: HCWs, healthcare workers

*"…I went to an ustadz, and he said that my disease was caused by witchcraft from my ex-husband."* (FGD, Dg. Turring, person affected by leprosy, female)

Another person affected mentioned that an *ustadz* refused the concept of *kandala*, i.e., as a consequence if the person was being dishonest or lying. Instead, they argued that leprosy was caused by sinful behaviors, such as engaging in sexual activity during menstruation.

*"According to an ustadz, the concept of 'kandala' does not exist. In Islam, as I understand it, leprosy results from the parents of the affected person who have had sex during menstruation. There are two possibilities of karma for the child: leprosy or disability."* (FGD, Dg. Ngiji, person affected by leprosy, female)

Further, leprosy was also explained as the result of one's parent's sins, for example committing acts of religious blasphemy, all linking back to the idea that kandala, and its consequences, was a punishment of sorts.

*"… the ustadz told me that I got leprosy because of blasphemy committed by my parents, because one is Christian, and one is Islam. So, I bear the punishment of my parents' sin."* (FGD, Dg. Firman, person affected by leprosy, male)

In addition to religious interpretations, perceptions were also linked with the belief that leprosy could be transmitted through sexual contact. This assumption may stem from the extended incubation period, which makes it challenging for individuals to trace the infection back to a specific exposure.

*"When I went for treatment, the HCW explained that leprosy could have been transmitted to me through sexual intercourse... In the past, I wasn't affected by leprosy; I developed leprosy only after I got married. The HCW said I got the infection from my husband."* (SSI, Dg. Ngasi, person affected by leprosy, female)

Or that transmission is related to hygiene, for example, spread through contact with animal feces.

*"… the second (cause of leprosy) is stepping on animal waste, cow's and buffalo's feces."* (SSI, Dg. Baji, person affected by leprosy, female)

In summary, people drew on cultural models that viewed leprosy as a result of religious taboos, sinful behavior, and/or moral failure, while also drawing on contagion models (i.e., infection from contact with an infected person or animal waste) and biological or hereditary models (i.e., infection is from parents at birth).

**Care seeking and treatment.** Participants showed limited knowledge about the early symptoms of leprosy: only 21 participants (5.2%) provided the correct responses with the lowest responses among community members, followed by close contacts. Apart from HCWs, loss of sensation was a less known symptom than skin lesions.

*"There was a patch. I thought it was just a patch, so I didn't go and check it… It was small at the beginning, later it became bigger, so I went to check it at the healthcare center."* (SSI, Dg. Naba, person affected by leprosy, male)

Half of the of the community members (75/150) responded that they did not know the early symptoms of leprosy; 45/150 (30%) mentioned other symptoms like disabilities and wounds (Table 3). The community members associated leprosy with disabilities, reflected in the following quote:

*"I don't really know [the symptoms of leprosy], but it looks like this (demonstrating a claw hand)."* (SSI, Dg. Sirajudding, community member, male)

When asked about their initial responses to the leprosy diagnosis, persons affected expressed feelings of shame, depression, fear of disability, and concerns about transmission, especially for their family members.

*"At that time, I wanted to die but it was not my time to die yet. I had this thought every single day."* (SSI, Dg. Baji, person affected by leprosy, female)

*"I was afraid, afraid that my child will be infected, but what can I do? My aunt had it, my husband had it, so I tried to keep my child away [to prevent the transmission]."* (SS1, Dg. Ngasi, person affected by leprosy, female)

A close contact shared that she felt pity for her son, who, after being diagnosed with leprosy, began isolating himself in his bedroom, especially when guests visited or when his friends came looking for him. Leprosy restricts those affected from socializing with neighbors and friends.

*"I feel pity for my son when his friends called him from outside (the house)... When his aunt came, she advised him not to sleep all day and not to stay in the bedroom all the time, which added more stress."* (SSI, Dg. Hayu, close contact, female)

While most participants correctly answered that leprosy is curable (309, 76.9%) and could be treated with medicine (244, 60.7%) (Table 3), leprosy treatment had multiple, often conflicting, meanings for people affected. The meanings were physiological (i.e., cure, prevent symptoms/disability, prevent transmission) coupled with social consequences. First, the meaning of treatment adherence and completion of required duration of treatment indicated biological cure.

*"The information about leprosy from the HCW was clear. If you get treatment immediately, leprosy can be cured. You need to take the medication regularly to get cured."* (FDG, Dg. Sopia, person affected by leprosy, female)

Adherence to treatment may alleviate symptoms associated with leprosy and prevent nerve damage and consequent disabilities. For example, a male participant mentioned he was afraid to become disabled in the absence of treatment, which might result in loss of his social network.

*"I felt bad. I had many thoughts, bad feelings, and fears [when he first went for leprosy treatment]. I was afraid that I would become disabled, and if that would happen that nobody would want to see me anymore… I tried to get treatment from a doctor. He said the medicine was available at the primary healthcare center. When I got the medication, I was no longer afraid."* (SSI, Dg. Naba, person affected by leprosy, male)

As pain due to nerve damage was a common symptom for persons affected with leprosy, the absence of pain was noted as cure. Even though this participant completed a year of treatment, he felt he was still not cured because he continued to experience pain.

*Interviewer: "Do you feel that your leprosy has been cured?"*

*Participant: "I still often feel pain on my feet. I have cramps. It is throbbing."*

*Interviewer: "Are you cured or not?"*

*Participant: "Not yet"* (SSI, Dg. Salehuddin, persons affected by leprosy, male)

Similarly, another meaning of cure was related to visual symptoms; when symptoms went away with medication (e.g., skin patches) they were cured.

*"I was told to take the medicine for 6 months, and during these 6 months the patches on my body started to disappear… and thank God, it was only 6 months that I was treated and cured."* (FGD, Dg. Sopia, person affected by leprosy, female)

While these definitions of cure are quite positive (i.e., lack of pain, lack of symptoms), accessing treatment potentially carried negative connotations in communities. The HCWs mentioned in interviews that the persons affected often felt ashamed if they were seen by other people visiting the leprosy clinics, especially those with disabilities or experiencing skin changes due to treatment, or when HCWs came to visit their homes for treatment.

*"… They (persons affected by leprosy) already understood that if they take medicine, they are no longer infectious. However, they are ashamed; especially those with disability, they usually don't want to go out… In such cases, I usually visit the patients at their house, but sometimes patients ask me not to come to their house. The reason is the neighbor may find out there is a leprosy patient in that house."* (SSI, Dg. Ngato, HCW, female)

Additionally, the treatment itself could also lead to adverse effects, such as skin discoloration, which is caused by clofazimine, one of the antibiotic drugs of multibacillary MDT. A female participant who was affected by paucibacillary leprosy explained how she did not experience skin discoloration from the treatment, which motivated her to continue treatment. However, this side effect is widely recognized and may discourage individuals from adhering to the treatment regimen.

*"At first, I was afraid that my skin would turn dark because it happened to my husband. However, when I took the medicine [MDT], there was no change in my skin color."* (SSI, Dg. Ngasi, person affected by leprosy, female)

Finally, treatment could reduce potential transmission. A female participant feared she would transmit leprosy to her child if she was not treated.

*"…I followed the HCW's advice [to take the MDT] because I was breastfeeding at that time, and I was afraid I would transmit the disease to my child…"* (SSI, Dg. Ngasi, person affected by leprosy, female)

In addition to biomedical treatment, participants also mentioned seeking advice from the ustadz and traditional healers to address their leprosy symptoms. Some recommended baths or drinks with a mixture of turmeric and jujube leaves (Ziziphus jujuba) – a traditional medicine and herbal remedy that is believed to have various health benefits, including anti-inflammatory and antioxidant properties.

*"…he [ustadz] told me to crush jujube leaves and turmeric, mix them with water, drink the mixture, and use the mixture for bathing."* (FGD, Dg. Turring, person affected by leprosy, female)

The people affected were also encouraged to engage in rituals and prayers of repentance in an effort to clean themselves of the condition, as leprosy in this context was often associated with impurity, sin, and witchcraft, often blaming the person affected.

*"I was told [by ustadz] to take a bath in the river. Later that night, we did a ritual to summon the spirits. I confessed that I will not commit the same sin (interreligious marriage, which according to the ustadz is the reason why this person has leprosy) as my parents did."* (FGD, Dg. Firman, person affected by leprosy, male)

Data from the survey showed similar findings: 44 community members (29.3%) and 13 HCWs (25.5%) expressed the belief that leprosy could be treated through these alternative methods (Table 3).

**Global Public Health**

### The complexities of leprosy-related stigma

As leprosy was historically associated with impurity and sin, and its name '*kandala*' carried negative connotations in this community, it is not unexpected that it carries complex and fluid forms of stigma. Quantitatively, the mean EMIC-CSS score (range 0–30) was 11.2 (SD 7.1). We found the community members had the highest stigma score (EMIC-CSS 11.7, 6.8), followed by HCWs (11.3, 6.2). The lowest score was among close contacts (10.6, 7.7; $p = 0.3$) (Table 1). Questions that were scored with higher stigma levels related to buying food from a person affected by leprosy, difficulty finding a job for a person with leprosy, disclosure of leprosy within the family, whether leprosy causes shame or embarrassment in the community and if a person with leprosy would try to keep others from knowing. The mean SDS score (range 0–21) was 8.7 (4.1). The highest SDS score was measured among the community members (9.5, 4.1), followed by close contacts (8.5, 4.0), and HCWs (7.3, 3.8; $p = 0.006$): those involved in leprosy care had lower social distance scores (5.7, 3.8) compared to HCWs not involved in leprosy care (8.0, 3.7; $p = 0.07$), although the difference was not significant (Table 1).

In the multivariate analysis, age < 30 or ≥50 years, completed secondary and tertiary education, being a self-employee, having temporary work/unemployment, a moderate income (95–189 USD/month) or higher income (≥190 USD/month) and HCWs or community members were associated with higher levels of perceived stigma (EMIC-CSS). For close contacts and community members, not having a relationship with a person affected by leprosy was related to higher EMIC-CSS scores. Determinants of higher SDS levels, i.e., maintaining higher levels of social distance towards affected individuals, were age < 30 years, HCW, self-employee, temporary work/unemployment or farmer, female, irregular income or lower income (<95 USD/month) or moderate income (95–189 USD/month) and community members (Table 2).

**Isolation and experience of stigma and discrimination.** Qualitatively, it was even more complex to unravel. In SSIs, we discussed aspects of stigma surrounding the everyday lives of persons affected in their homes, in their wider communities and within the healthcare system. Experiences of stigma infiltrated health system experiences and extended to HCWs working in the leprosy clinics as well (*see also next section*).

The decision to disclose one's leprosy status was influenced by fear of facing stigma or abandonment. Survey findings indicated that 45/75 individuals affected by leprosy (60.0%) chose not to disclose their condition to others, often because of concerns about potential community rejection. This is exemplified by the fact that 55/126 close contacts (44%) were unaware of their relationship with someone who had leprosy. Further, 19/75 people (25.3%) felt the community would have less respect for them because of their illness. In addition, 10/75 participants (13.3%) acknowledged that other people refused to visit their house because of leprosy, even after treatment, and 9/75 (12.0%) stayed away from work or social gatherings. These findings were confirmed during the interviews: persons affected by leprosy concealed their condition for various reasons, including apprehension of stigma or discrimination, feelings of shame, and concerns about livelihood. Even within the clinic, there were instances where HCWs engaged in subtle stigmatizing practices, like whispering results to individuals diagnosed with leprosy, perpetuating secrecy and discouraging disclosure.

Historically, leprosy is associated with quarantine and isolation in leprosy settlements. The perception was that persons affected should avoid social interactions and stay at home. This was experienced by leprosy-affected participants:

*"I had leprosy myself. I know that people affected by leprosy feel isolated, intimidated, avoided, and are kept away from or separated from their family and community. People affected by leprosy feel an extraordinary sense of "depression."* (FGD, Dg. Bilal, person affected by leprosy, male)

But this was also the perception of families with members who had leprosy.

*"No one knows about my son's disease. I never told anyone about it, neither our family nor the community. … He hides in his bedroom. When I was asked about my son, I would say he was sleeping."* (SSI, Dg. Hayu, close contact, female)

Beyond the isolation, another perception experienced by those affected with leprosy is that they are shunned.

*"The only one who knew about my condition was my husband because he had leprosy as well. My neighbors and my children didn't know. I hid everything. When they asked what this was on my face, I said allergy… People with leprosy are shunned; other people are disgusted. I am a coconut shredder; if they knew that I had leprosy, all my customers would run away. There would be no livelihood for me."* (SSI, Dg. Baji, person affected by leprosy, female)

This led some persons affected to hide the symptoms of leprosy or leprosy treatment, like visible skin changes, i.e., hyperpigmentation resulting from treatment. They would refer to these changes as an allergic reaction, cover their face with a mask or hijab (for women), or cover their skin by wearing long sleeves.

*"At the beginning, I wore a mask when I was in my workplace because of the redness on my face."* (FGD, Dg. Bilal, person affected by leprosy, male)

Persons affected by leprosy expressed during the interviews that they faced stigma and discrimination in their daily lives. From their perspective, community members sometimes avoid physical contact (e.g., handshakes, touching objects handled by persons affected) and/or maintained a physical distance. This could make people feel shunned.

*"At school, we greet the teachers with a handshake. There is one teacher, every time I want to shake his hand, he pulls his hand away."* (FGD, Dg. Fajrin, person affected by leprosy, male)

But it was not just from the community, the social exclusion was integrated into the family as well, again reaffirming the need to avoid disclosure.

*"My mom said that if people knew I had leprosy, they would be disgusted. At that time, none of my family members knew about my condition; I spoke up and revealed my condition, but afterwards I was treated differently (crying) … When I visited my auntie, she would wipe the bed and cover it with a cloth when I sat on it."* (FGD, Dg. Manohara, person affected by leprosy, female)

As a result, affected individuals were often excluded from social activities. The lack of knowledge regarding leprosy and its transmission among the community contributed to social exclusion.

*"I was almost expelled from school because of leprosy. At the time, when I was diagnosed with leprosy, we had online school because of COVID-19. After school returned to in-person learning, I was told to remain in the online class because they feared I would transmit the disease to other students. A HCW helped me, spoke to the headmaster and showed a letter stating that I had received treatment, and was no longer infectious."* (FGD, Dg. Firzan, person affected by leprosy, male)

In addition, social exclusion was manifested as difficulties in securing employment and/or rejection to participate in community activities. For example, participants relayed how hotel staff prohibited people affected by leprosy to organize a gathering at a hotel.

*"I have noticed that persons affected by leprosy have more difficulties finding a job if people (employers) know they have leprosy."* (SSI, Dg. Sirajudding, community member, male)

*"We even had an event at a hotel and invited our friends who are also affected by leprosy. We were asked to leave the premises because of leprosy. We were prohibited from participating in any activities at the hotel."* (FGD, Dg. Ngato, person affected by leprosy, female)

**The consequences of elimination.** In 2000, Indonesia formally reached the WHO elimination status, after which the leprosy control program underwent significant downscaling on the health system front. This led to loss of clinical expertise, resources and political commitment dedicated to leprosy control efforts [17].

Participants spoke about how, at times, HCWs in charge of the leprosy program were not available at the clinics when persons affected by leprosy, or their families, travelled to the clinic to pick up their drugs. As a result, participants experienced feelings of insecurity when considering visiting the clinic. They were hesitant to be seen at the leprosy clinic and were concerned about disclosing their leprosy status to HCWs who were not affiliated with the leprosy program.

*"I didn't know what disease this was, but when I knew it wasn't good, I felt hurt. When I went to fetch the medicine in the healthcare center, because many people who work there know me, so when I went there, I fetched the medicine secretly, because I was very much ashamed [….] Just HCWs from Leprosy [program] know …. No one knows I was sick, I don't know if a nurse sees my data in the healthcare center, but it has been a long time ago, and if it is a long time ago it is stored in the cupboard, so no way the nurse knows, it is not their field."* (SSI, Dg. Baji, person affected by leprosy, female)

In interviews, the leprosy program HCWs reported that they received substantial lower annual funding compared to priority programs such as tuberculosis or childhood stunting. The implications of limited budget allocation were substantial, as efforts toward leprosy elimination—such as contact tracing and community examinations—were no longer conducted optimally.

*"The fund for the leprosy program at our [primary healthcare] clinic is at most 6 million rupiah (~USD 370) per year. The funding can only cover 10 contact tracing activities in the community. How can this fund cover a total of 52,000 population [within the working area]? For the tuberculosis program, the allocated funding is 40 million rupiah (~USD 2,465) per year."* (SSI, Dg. Ngato, HCW, female)

*"There were many activities that we had planned but not approved because of lack of funding. Currently, the funding for leprosy is between 2.5 and 5 million rupiah per year (equivalent to USD 154-308). Prior to the COVID-19 pandemic, we were allocated 20 million rupiah annually for the leprosy program (~USD 1,233). However, due to the shift in priorities towards COVID-19, the funding for the leprosy program was reduced. Currently, the most prioritized national health program is stunting, therefore it receives more funds."* (SSI, Dg. Sibali, HCW, male)

HCWs involved in leprosy care received only basic training with limited opportunities for refresher courses. Those who were not directly engaged in leprosy care received no training, which impacted the quality of care for patients.

*"I received my first and last basic leprosy training in 2005. The training provided an introduction to leprosy, the treatment, as well as communication techniques. There was a training related to self-care (of leprosy patients) for three days in 2015"* (SSI, Dg. Ngato, HCW, female)

Interviewer: "Have you been exposed to any leprosy training since working here (as a nurse in the emergency ward since 1986)?"
*Participant: "No, I haven't."* (SSI, Dg. Hamsina, HCW, female)

The lack of training resulted in insufficient knowledge about leprosy. The interviewed HCWs stated that some HCWs were reluctant to physically examine or treat individuals with leprosy who had wounds.

*"Frankly speaking, HCWs rarely understand leprosy, except those who have been trained on leprosy. Many doctors do not understand the disease as well, especially the newly graduated ones. They do not understand the transmission and symptoms of leprosy... There was a doctor who didn't want to touch a leprosy patient…"* (SSI, Dg. Ngato, HCW, female)

*"From my point of view, HCWs should have no objection to treat wounds of persons affected by leprosy. That's their job. However, there are some who don't like to do it; they are disgusted by the wounds."* (SSI, Dg. Hamsina, HCW, female)

## Discussion

Overall, our findings revealed significant gaps in knowledge and complex forms of stigma, both of which impact leprosy public health efforts and eventual eradication of leprosy. We first discuss the experiences of persons affected and their families and communities, followed by considerations related to healthcare workers and health systems.

First, survey results revealed poor knowledge on leprosy, particularly the etiology, transmission and symptoms, among persons affected, close contacts, and community members. In the SSIs and FGDs, however, we found that there were multiple narratives used to explain leprosy, its transmission, and treatments. Other researchers echoed similar community-derived explanations, shaped by both local and global belief systems and cultural influences [18,19]. For example, a study in India and Indonesia found that certain types of food, e.g., unhealthy food or seafood, and an unclean environment could cause leprosy. Similar to our findings, a study from another region of Indonesia, found the belief that conception occurring during sexual intercourse while menstruating could result in a child developing leprosy [7]. We also explored the local terminologies used (i.e., *kandala*) and the negative impact these words have on disease perception in communities. Researchers in Niger conducted a study focused on semiotics and stigma related to leprosy and highlighted the value of understanding language as a way to better understand the complexities of stigma but also helped them to tailor interventions using the terminologies that are preferred by those with lived experience [7].

Second, stigma associated with leprosy was complex, occurred in a variety of social processes and with a wide range of actors, and often overlapped with aspects of the leprosy narratives (e.g., leprosy is a result of sin). Studies across various religious contexts have shown that individuals affected by leprosy are often viewed as receiving a punishment from higher powers or facing consequences for perceived sins [20,21]. This belief is not unique to a specific region, as evidenced by another study conducted in Indonesia [7], which highlighted a religious belief within Islam linking leprosy to engaging in sexual intercourse during menstruation. Across various religions and cultures, menstruation is associated with notions of impurity, leading to practices of isolation, exclusion from religious and educational activities, and restrictions on sexual relations [22–24].

These beliefs and practices have been formed over centuries and continue to shape community attitudes towards leprosy. Beyond religion, scholars have also argued, in this case in India, the role of public health messaging and historical colonization practices in perpetuating leprosy-related stigma and placing the blame on affected individuals [18]. While there have been successful interventions documented in India [25] and Indonesia that positively impacted community perceptions and knowledge about leprosy, there remains a need for more comprehensive ethnographic and linguistic data in our context. Such data will be crucial to inform the development of effective community-based interventions aimed at addressing leprosy-related stigma.

Finally, in our study, HCWs had a mean knowledge score of 5.0 (SD 1.7), falling below the expected standards for healthcare professionals. This deficiency was particularly evident in their understanding of the early symptoms, cause and transmission of leprosy, as half or even less than half of HCWs provided the correct answers. Inadequate knowledge on leprosy may can have negative implications on the quality of care delivered to persons affected by leprosy. It may result in misdiagnosis, treatment delays and improper management, increasing the risk of developing disabilities and exacerbating stigma and social exclusion [26,27]. These factors may further marginalize affected individuals and hinder their access to necessary healthcare services. These findings are consistent with previous studies conducted in India and Indonesia [15]. Authors of a systematic literature review argued that in order to enhance HCWs' perceptions and knowledge of leprosy, structural investments are needed by governments and related stakeholders to improve leprosy training programs [27]. However, given the constraints of reduced funding and healthcare capacity, as described by our participants, this target may prove to be challenging. As these data were collected alongside the MetLep trial, we have been able to implement engagement activities with the community and healthcare workers to promote the study and utilize the study as a platform to facilitate discussions about leprosy.

This study has three limitations. First, in the quantitative component of this study, the majority of participants were women (76%). This may reflect their greater availability for daytime data collection, as majority were engaged in household work, and may not fully represent the broader population. To address this limitation, the qualitative section sought, where possible, to recruit a more balanced proportion of men and women. Second, one of the KAP survey questions addressed leprosy prevention. This item was adapted from earlier KAP studies conducted in areas where post-exposure prophylaxis (PEP) for leprosy was introduced. Since PEP has not yet been implemented in South Sulawesi, participants may have been unfamiliar with this relatively new intervention. Third, this study did not collect data on participants' self-identification regarding gender. In Bugis-Makassar communities, five gender identities are recognized (as described in contemporary anthropological literature), each associated with distinct social roles and expectations that may influence the experience of living with leprosy [28]. We recommend that future research incorporate context-specific considerations (e.g., through the use of rapid ethnographic tools to capture such dimensions) prior to finalizing survey instruments.

Knowledge, attitudes and practices regarding leprosy have been extensively studied, with determinants of knowledge and community stigma to vary across local contexts and communities. Our findings exposed limited knowledge of leprosy across all four participant groups, alongside high levels of stigma and a strong desire to maintain distance from affected individuals. Addressing these issues requires targeted interventions, but compels a thorough understanding of the historical, social and cultural dimensions of leprosy in that specific context.

Culturally sensitive educational programs that provide accurate information on leprosy, its transmission, and treatment can help to demystify the disease and reduce fear. Equally, it is critical to ensure that public health programmes do not inadvertently perpetuate stigma within communities or the healthcare system. Such initiatives should ideally form part of a comprehensive approach, integrating leprosy education into broader public health campaigns.

Strengthening leprosy control efforts also requires regular training and refresher courses for HCWs to improve diagnosis, contact tracing, community engagement, and stigma reduction. Achieving this, however, depends on the prioritization of national leprosy programs, with sufficient funding and sustained political commitment. This is particularly important at a time where the pursuit of leprosy elimination goals dominates political and public health agendas, often diverting resources from broader, long-term priorities, such as health system strengthening, community engagement and stigma reduction.

## Supporting information

**S1 Text. The detailed description of the model explaining the relationship between the dependent and independent variables using a Bayesian approach.**
(DOCX)

## Acknowledgments

We wish to thank Anna van 't Noordende and Wim van Brakel for their technical advice in developing this study. We extend our sincere gratitude to all individuals who participated in this study.

## Author contributions

**Conceptualization:** Ralalicia Limato, Sri V. Muchtar, Hardyanto Soebono, Jennifer I. Van Nuil, Marlous L. Grijsen.

**Data curation:** Ralalicia Limato, Jennifer I. Van Nuil, Marlous L. Grijsen.

**Formal analysis:** Ralalicia Limato, Ida A. Sutrisni, Rahmat Sagara, Asyhad F. Abdillah, Iqbal Elyazar, Jennifer I. Van Nuil, Marlous L. Grijsen.

**Funding acquisition:** Marlous L. Grijsen.

**Investigation:** Ralalicia Limato, Ida A. Sutrisni, Yuliati Yuliati, Al Kadri, Sri V. Muchtar, Iqbal Elyazar, Hardyanto Soebono, Jennifer I. Van Nuil, Marlous L. Grijsen.

**Methodology:** Ralalicia Limato, Ida A. Sutrisni, Rahmat Sagara, Asyhad F. Abdillah, Sri V. Muchtar, Iqbal Elyazar, Hardyanto Soebono, Jennifer I. Van Nuil, Marlous L. Grijsen.

**Project administration:** Ralalicia Limato, Marlous L. Grijsen.

**Resources:** Marlous L. Grijsen.

**Software:** Rahmat Sagara, Asyhad F. Abdillah, Iqbal Elyazar, Jennifer I. Van Nuil.

**Supervision:** Yuliati Yuliati, Al Kadri, Sri V. Muchtar, Iqbal Elyazar, Hardyanto Soebono, Marlous L. Grijsen.

**Validation:** Ralalicia Limato, Rahmat Sagara, Asyhad F. Abdillah, Jennifer I. Van Nuil, Marlous L. Grijsen.

**Writing – original draft:** Ralalicia Limato, Ida A. Sutrisni, Jennifer I. Van Nuil, Marlous L. Grijsen.

**Writing – review & editing:** Ralalicia Limato, Ida A. Sutrisni, Jennifer I. Van Nuil, Marlous L. Grijsen.

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
