## [Decision Letter · Decision Letter 0]

18 Jul 2025

PGPH-D-25-00556

Exploring leprosy perceptions in South Sulawesi, Indonesia: a mix-method study on knowledge, attitudes, practices, and stigma

Dear Dr. Grijsen,

Thank you for submitting your manuscript to PLOS Global Public Health. After careful consideration, we feel that it has merit but does not fully meet PLOS Global Public Health’s publication criteria as it currently stands. Therefore, we invite you to submit a revised version of the manuscript that addresses the points raised during the review process.

Please address all comments from reviewers in order to be considered for publication. Note that Reviewer 2 has uploaded their comments as an attachment. 

We look forward to receiving your revised manuscript.

Kind regards,

Jessica K Fairley, MD, MPH

Academic Editor

Journal Requirements:

1. Please ensure that your Ethics Statement is available in its entirety at the beginning of your Methods section, under a subheading 'Ethics Statement'.

2. Please provide separate figure files in .tif or .eps format.

3. Tables should not be uploaded as individual files. Please remove these files and include the Tables in your manuscript file as editable, cell-based objects. For more information about how to format tables, see our guidelines: 

https://journals.plos.org/globalpublichealth/s/tables

4. We have noticed that you have uploaded Supporting Information files, but you have not included a list of legends. Please add a full list of legends for your Supporting Information files after the references list.

5. In the online submission form, you indicated that “Data will be made available upon reasonable request.”. 

3. Uploaded as supplementary information.

Additional Editor Comments (if provided):

Reviewers' comments:

Reviewer's Responses to Questions

**Comments to the Author**

1. Does this manuscript meet PLOS Global Public Health’s publication criteria?

Reviewer #1: Yes

Reviewer #2: Yes

2. Has the statistical analysis been performed appropriately and rigorously?

Reviewer #1: Yes

Reviewer #2: Yes

3. Have the authors made all data underlying the findings in their manuscript fully available (please refer to the Data Availability Statement at the start of the manuscript PDF file)?

Reviewer #1: No

Reviewer #2: Yes

4. Is the manuscript presented in an intelligible fashion and written in standard English?

Reviewer #1: Yes

Reviewer #2: Yes

Reviewer #1: This was a well-constructed study, and the data is very clearly presented. I think it is important in showing that leprosy stigma and lack of knowledge, particularly about the disease in its early symptoms, continue to have a significant impact on people's lives in Sulawesi, despite Indonesia's status as having "eliminated" the disease. I liked that the authors suggest some of the complex reasons why people might have certain beliefs. I agree that the long incubation period is one of the reasons people in many parts of the world speculate on ways they might have been exposed, often leading to misconceptions.

In terms of the belief that leprosy is hereditary, it is also easy to see why people might think this, as leprosy susceptibility might be inherited—you could point out that household contacts (often family members) are at a higher risk, and this, coupled with possible susceptibility running in families, could make it seems like leprosy itself could be passed down from parent to child. The belief that one of your participants discussed of the possibility of leprosy transmission through livestock feces is one that is the subject of a few ongoing studies currently.

In designing demographic questions, I think it is important for researchers to be aware of gender diversity rather than sticking to the gender binary (male/female) to have participants self-identify. This is perhaps even more significant since the Bugis-Makassar are known (at least in the contemporary anthropological literature) as recognizing five gender identities, and with these different identities come different roles and expectations that also might play a part in the experience of having leprosy. I’m not sure if you have this data on whether people identified with the gender/sex assigned at birth or not, but if not, this could be something to consider for future studies. Sharyn Graham-Davies has written extensively on this, but this is her book that focuses on the Bugis culture.

Davies, Sharyn Graham. "Challenging Gender Norms: Five Genders Among Bugis in Indonesia (Case Studies in Cultural Anthropology) by Graham Davies, Sharyn (2006) Paperback. I know this text might not be readily accessible, but this summarizes some of her research https://www.bbc.com/travel/article/20210411-asias-isle-of-five-separate-genders

This is just a suggestion, but I think it can humanize participants you are quoting to use pseudonyms rather than numbers. If you choose to use pseudonyms, you can use typical local names for the age of the participants.

I just have a few more minor notes by line of the manuscript:

Line 77-78: “The impact of leprosy and the associated disabilities has taken a multifaceted toll on the physical, psychological, social, and economic aspects of those affected.” Suggested rewording: “Leprosy at the associated disabilities often have a negative impact on the physical, psychological, social, and economic aspects of the lives of people affected.”

Line 99: “from Bugis-Makassar ethnic”—maybe “from the Bugis-Makassar ethnic groups”

Line 244: You might include the different word origins. You mention kandala as a Bugis-Makassar specific term, but I think kusta comes from Sanskrit and Hindu influence in Indonesia.

Line 278: Maybe put “leprosy of the skin” and “leprosy of the bones” in quotation marks here.

Reviewer #2: This article is interesting because leprosy is still a health problem in several countries. Public knowledge, stigma and discrimination seem to be urgent matters to be addressed in government programs. Research findings to support program development need to be carried out and presented as a reference. If you agree, I have attached some suggestions regarding this article

**Do you want your identity to be public for this peer review?** For information about this choice, including consent withdrawal, please see our Privacy Policy

Reviewer #1: **Yes: ** Cassandra White

Reviewer #2: No

---

## [Decision Letter · Decision Letter 1]

4 Dec 2025

Exploring leprosy perceptions in South Sulawesi, Indonesia: a mixed-methods study on knowledge, attitudes, practices, and stigma

PGPH-D-25-00556R1

Dear Dr. Grijsen,

We are pleased to inform you that your manuscript 'Exploring leprosy perceptions in South Sulawesi, Indonesia: a mixed-methods study on knowledge, attitudes, practices, and stigma' has been provisionally accepted for publication in PLOS Global Public Health.

Best regards,

Julia Robinson

Executive Editor

Reviewer Comments (if any, and for reference):

Reviewer's Responses to Questions

**Comments to the Author**

Reviewer #1: All comments have been addressed

publication criteria?

Reviewer #1: Yes

3. Has the statistical analysis been performed appropriately and rigorously?

Reviewer #1: (No Response)

4. Have the authors made all data underlying the findings in their manuscript fully available (please refer to the Data Availability Statement at the start of the manuscript PDF file)?

Reviewer #1: Yes

5. Is the manuscript presented in an intelligible fashion and written in standard English?

Reviewer #1: Yes

Reviewer #1: I think this is an important contribution to the literature on the complexities of leprosy stigma in the 21st century in a country (Indonesia) where it has been "eliminated" but where many people are still affected by the disease and its stigma. The perspectives from different groups, including people affected, healthcare workers, close contacts, and community members, including interview excerpts, provide more depth to the article, and I appreciated that the authors took the time to replace participant numbers with pseudonyms in this version, which I also think helps readers think about participants in different ways than numbers do.

There are some significant points that come from this paper, particularly related to lack of knowledge in the community and often among healthcare workers about leprosy. I can’t remember if I noticed this on the first read, but the fact that many people (55/126 of close contacts “were unaware of their relationship with someone who had leprosy” says so much about the tricky problem of disclosure vs. nondisclosure; if more people disclose the disease, there might be more knowledge and normalization, but the risk of stigmatization still seems too strong for most people to want to do this.

**Do you want your identity to be public for this peer review?** For information about this choice, including consent withdrawal, please see our Privacy Policy

Reviewer #1: **Yes: ** Cassandra White
